# Clinical Implication of Liquid Biopsy in Colorectal Cancer Patients Treated with Metastasectomy

**DOI:** 10.3390/cancers13092231

**Published:** 2021-05-06

**Authors:** Soohyeon Lee, Young-Soo Park, Won-Jin Chang, Jung Yoon Choi, Ahreum Lim, Boyeon Kim, Saet-Byeol Lee, Jong-Won Lee, Seon-Hahn Kim, Jin Kim, Jung-Myun Kwak, Kyung-Chul Yoon, Sung-Ho Lee, Yeul Hong Kim

**Affiliations:** 1Division of Medical Oncology and Hematology, Department of Internal Medicine, Korea University College of Medicine, Seoul 02841, Korea; soohyeon_lee@korea.ac.kr (S.L.); hippo310@korea.ac.kr (W.-J.C.); chkmg1@korea.ac.kr (J.Y.C.); ahreumlim@korea.ac.kr (A.L.); 2Korea University Cancer Research Institute, Korea University College of Medicine, Seoul 02841, Korea; difco@korea.ac.kr (Y.-S.P.); alice_1989@korea.ac.kr (B.K.); akdltmxkf@korea.ac.kr (S.-B.L.); potato23@korea.ac.kr (J.-W.L.); 3Department of Surgery, Korea University College of Medicine, Seoul 02841, Korea; drkimsh@korea.ac.kr (S.-H.K.); mrgs@korea.ac.kr (J.K.); jmkwak@korea.ac.kr (J.-M.K.); kcyoon123@snu.ac.kr (K.-C.Y.); 4Department of Thoracic and Cardiovascular Surgery, Korea University College of Medicine, Seoul 02841, Korea; sholeemd@korea.ac.kr

**Keywords:** circulating tumor DNA, liquid biopsy, metastatic colorectal cancer, metastasectomy

## Abstract

**Simple Summary:**

Circulating tumor DNA (ctDNA) is tumor-derived fragmented DNA in the bloodstream that was shed from primary and/or metastatic tumors. ctDNA in patients with oligometastatic colorectal cancer (CRC) was detected before and after metastasectomy depending on the clinical context. The detection rate of ctDNA was higher in liver metastasis than lung metastasis and tumors measuring ≥1 cm and small tumors <1 cm. After metastasectomy for oligometastatic lesions with good response to neoadjuvant chemotherapy, most ctDNA was cleared or existed below the detection level. Biological characteristics affecting tumor DNA release should be considered when applying ctDNA assays in clinical settings.

**Abstract:**

Background & Aims: The application of circulating tumor DNA (ctDNA) has been studied for predicting recurrent disease after surgery and treatment response during systemic treatment. Metastasectomy can be curative for well-selected patients with metastatic colorectal cancer (mCRC). This prospective study investigated the ctDNA level before and after metastasectomy in patients with mCRC to explore its potential as a predictive biomarker. Methods: We collected data on 98 metastasectomies for mCRC performed from March 2017 to February 2020. Somatic mutations in the primary and metastatic tumors were identified and tumor-informed ctDNAs were selected by ultra-deep targeted sequencing. Plasma samples were mandatorily collected before and 3–4 weeks after metastasectomy and serially, if patients agreed. Results: Data on 67 of 98 metastasectomies (58 patients) meeting the criteria were collected. ctDNA was detected in 9 (29%) of 31 cases treated with upfront metastasectomy and in 7 (19.4%) of 36 cases treated with metastasectomy after upfront chemotherapy. The detection rate of ctDNA was higher in liver metastasis (*p* = 0.0045) and tumors measuring ≥1 cm (*p* = 0.0183). ctDNA was less likely to be detected if the response to chemotherapy was good. After metastasectomy, ctDNA was found in 4 (6%) cases with rapid progressive disease. Conclusion: The biological factors affecting the ctDNA shedding from the tumor should be considered when applying ctDNA assays in a clinical setting. After metastasectomy for oligometastatic lesions in good responders of chemotherapy, most ctDNA was cleared or existed below the detection level. To assist clinical decision making after metastasectomy for mCRC using ctDNA, further studies for improving specific outcomes are needed.

## 1. Introduction

Colorectal cancer (CRC) is the third most common cancer in the world and Korea [1]. Approximately 25% patients with CRC present with overt metastasis and 25–35% patients develop metastasis during the course of the disease [2]. The majority of these patients have a poor prognosis. However, some patients with metastatic CRC (mCRC) are diagnosed with isolated liver and lung metastases. Surgical resection is the treatment of choice, with a survival rate of 25–50% [3]. Metastasectomy can be performed if the liver or lung volume after resection is sufficient to maintain organ function while removing all macroscopic disease with microscopically negative margins and preserving adequate vascular inflow and outflow [4]. Patients with widely metastatic disease may be suitable candidates for local treatment depending on the systemic chemotherapy response. mCRC treatment strategies aim at converting irresectable disease into resectable disease. The 5-year survival after conversion ranges from 35% to 50% [5,6], which is similar to that for patients who underwent resection at presentation.

Although the survival of patients with resectable stage IV cancer has significantly improved with effective chemotherapy regimens and advances in surgical techniques, 75% patients who undergo metastasectomy developed recurrence within 18 months of surgery [7]. Moreover, there is no consensus on standard treatment guidelines regarding the role of postoperative chemotherapy, treatment period, and additional intervention for preventing recurrence after metastasectomy. In addition, difficulty in detecting tiny lesions during postmetastasectomy follow-up, lack of reliable longitudinal surveillance methods, and multiple genetic changes because of clonal evolution and corresponding intratumoral heterogeneity are limitations for customizing treatment strategies to mCRC.

Liquid biopsy has emerged as a novel method for tumor mutation profiling. Various tumor-derived products such as circulating tumor cells, circulating cell-free DNA, and circulating tumor DNA (ctDNA) can be detected in the blood, which has been increasingly used in clinical practice [8]. The DNA fragments from tumor cells are released into the circulation through apoptosis, necrosis, and secretion. Furthermore, tumor-specific genetic alterations such as driver mutations, chromosome copy number alterations, and methylation can be detected in ctDNA, which can be of high value for cancer detection, prognostication, and treatment monitoring [9]. Several studies have confirmed the predictive value of ctDNA levels in mCRC and the prognostic significance of postoperative ctDNA levels in early and locally advanced CRC [10,11,12,13,14]. The presence of ctDNA after radical surgery correlates with recurrence, and elevated ctDNA levels in R0 and R1 resections may signal the occurrence of micrometastases [15,16,17]. 

In this study, we analyzed tumor-specific DNA mutations found in primary and metastatic tumor tissues and investigated the presence of tumor-guided ctDNA in patients with oligometastatic CRC before and after metastasectomy in a clinical setting and explored its practical usefulness.

## 2. Patients and Methods

### 2.1. Study Design and Participants

We prospectively enrolled patients with mCRC who underwent surgical resection of the primary tumor and metastasectomy at Korea University Anam Hospital between March 2017 and February 2020 with the objective of assessing premetastasectomy and postmetastasectomy ctDNA levels and clinical outcomes. Patients aged >19 years with stage Ⅳ CRC who underwent metastasectomy with definitive intent and with or without systemic chemotherapy were included. The decision regarding on chemotherapy timing and regimen was made by the treating physicians based on the current treatment guidelines. The treating physicians were blinded to the ctDNA results. All participants’ samples were collected with informed consent. This study was approved by the Institutional Review Board of Korea University Anam Hospital (2017AN0070).

### 2.2. Tumor Tissue and Blood Collection and Mutation Analysis by Next-Generation Sequencing

All primary and metastatic tumor samples were obtained from formalin-fixed paraffin-embedded surgical specimens based on a cutoff value of 80% for tumor sample purity. The resected primary and metastatic tumor specimens were evaluated by next-generation sequencing (average depth × 2328; on-target rate: 96.61%; and average uniformity: 97.69%) using the Ion AmpliSeq Cancer Hotspot Panel (ICP) v2 with the Ion Torrent Proton system. The ICP v2 covers 2800 COSMIC mutations in 50 cancer-related genes, which are presented in Appendix A. The mutational status of tumor tissues was determined as previously described [18].

Blood was drawn from patients with mCRC before and 3–4 weeks after metastasectomy. Serial blood sampling after metastasectomy was conducted if the patients agreed. At each time point, 10 mL of blood was collected in EDTA tubes and processed for 1 h in the circulating cancer biomarker laboratory in the study institution. Ultra-deep targeted sequencing of ctDNA (average depth × 25,804; on-target rate: 90.18%; and average uniformity: 97.89%) was performed using the ICP v2 for paired plasma samples, as reported previously [19].

### 2.3. Tumor-Guided ctDNA Identification

We developed a personalized approach for tumor-guided ctDNA detection and quantification called targeted sequencing. We selected pathogenic somatic variants in primary and metastatic CRC tumor samples and then matched plasma sample with same pathogenic variants from tumor cells. Pathogenic variants were annotated as “likely pathogenic” or “pathogenic” in the COSMIC, ClinVar, and OncoKB databases. Each alteration identified de novo was reviewed manually to exclude false positives. The cutoffs of variant allele frequency (VAF) for tissue sequencing and ctDNA sequencing were 5% and 1%, respectively, if the mutation was identified in the tumor specimen. Patients with a VAF below these thresholds were considered negative, whereas those with missing VAF data for a mutation were excluded from further analysis. We selected tumor-ctDNA matches and comprehensive filtering of mutations through clustering maximized the likelihood that the reported alterations originated from tumors and not from other sources such as clonal hematopoiesis of indeterminate potential [20]. The germline mutations were not analyzed in this study and were filtered when plasma variants uniformly represented in a VAF of 50–100%.

### 2.4. Statistical Analysis

Descriptive statistics for categorical variables are presented as frequencies and those for continuous variables are presented as means. Categorical variables are presented using contingency tables. They were compared using the chi-square test or Fisher’s exact test. The ctDNA level was classified as detectable (ctDNA positive, VAF: ≥1%) or undetectable (ctDNA negative, VAF: <1%). The relationships between clinicopathological variables and ctDNA detection and VAF levels were assessed using Fisher’s exact test and the Wilcoxon rank-sum tests. Progression-free survival (PFS) was measured from the time of metastasectomy to recurrence or death from disease. Patients alive at the last follow-up and with no evidence of disease were censored. Overall survival (OS) was measured from the time of metastasectomy to death from the disease. *p* < 0.05 was considered statistically significant. All statistical analyses were performed using SPSS ver. 25.0 (IBM Corp., Armonk, NY, USA).

## 3. Results

### 3.1. Patient Characteristics

This study enrolled consecutive patients with mCRC who underwent metastasectomy with or without chemotherapy. Table 1 summarizes the preoperative features of the 58 evaluable patients. The mean age of the patients was 56 years, and two-thirds of the enrolled patients were male (*n* = 37, 63.8%). The distribution of the primary tumor sites was as follows: the ascending colon (*n* = 8, 13.8%), descending colon (*n* = 20, 34.5%), and rectum (*n* = 30, 51.7%). In all patients, except one, the primary lesion was surgically removed. At the time of diagnosis, metachronous metastasis occurred in one-third of patients and synchronous metastasis occurred in two-thirds of patients. Metastasectomy was most commonly performed for liver metastasis (66%), followed by lung (43%), peritoneal (7%), and lymph node (3%) metastases.

### 3.2. ctDNA Detection before and after Metastasectomy

Data on 98 metastasectomies were collected; however, data on 31 of 98 metastasectomies (3 cases showing no cancer cells after metastasectomy, 10 cases that failed quality control during the sample preparation process, and 18 cases lacking pathogenic variants in the metastatic tumor tissue; Figure 1) were excluded. The clinical data on every metastasectomy and pathogenic variants in ctDNA are presented in Appendix A. Overall, ctDNA was detected in 16 of 67 (24%) cases before metastasectomy and in 4 of 67 (6%) cases with rapid progressive disease immediately after metastasectomy.

In the clinical context, metastasectomy of mCRC can be performed in 2 situations. First, if the metastatic lesion is isolated and small, upfront metastasectomy without systemic chemotherapy can be performed (groups 1 and 2). Second, if the metastatic disease is extensive and not resectable, systemic chemotherapy is preferred and, thereafter, the decision to perform metastasectomy can be reassessed depending on the chemotherapy response (groups 3 and 4).

### 3.3. Group 1: Upfront Metastasectomy without Neoadjuvant Chemotherapy (R0 Resection = 28 Cases)

Patients in this group underwent metastasectomy without neoadjuvant chemotherapy (31 of 67 cases). R0 resection failed in 3 cases. The majority of 28 successful R0 resection cases were found to be oligometastatic and metachronous (26 cases). ctDNA was detected in 8 (28.5%) of 28 R0 resection cases before metastasectomy, but it was not detected in any case after metastasectomy (Table 2). All 8 cases positive for ctDNA were cases of liver metastasis and the tumor size was >1 cm. The VAF in primary tumor was often not detected (case number #2, #34, #68) when the primary tumor was exposed to chemotherapy or concurrent chemoradiotherapy in metachronous mCRC (26 cases). The detection of ctDNA before metastasectomy was not related to PFS and OS. 

### 3.4. Group 2: Upfront Metastasectomy without Neoadjuvant Chemotherapy (R0 Resection Failed = 3 Cases)

Patients in this group underwent metastasectomy without neoadjuvant chemotherapy. ctDNA was detected in 1 (33.3%) of the 3 R0 resection cases. One ctDNA positive case had liver metastasis with a 2.1 cm sized tumor. Although residual lesions in the lymph node, peritoneum, and lung (small sized multiple nodules) were found on postoperative imaging, ctDNA was not detected immediately after metastasectomy (Table 3). 

We compared the median VAFs in the tumor tissue and ctDNA between groups 1 and 2. The amount of ctDNA released from the tumor can be considered to reflect the tumor burden if the size of metastatic lesions is large and chemotherapy is not performed. The median VAF in the tumor was 28.7% and that in ctDNA was 1.1%, meaning that the detection rate of ctDNA was less than one-tenth of that of tumor. 

### 3.5. Group 3: Neoadjuvant Chemotherapy Followed by Metastasectomy (R0 Resection = 25 Cases) 

This group of patients received upfront chemotherapy because curative surgery was not feasible at the time of mCRC diagnosis due to metastatic tumor burden and extent. Most cases (22 of 25 cases) had synchronous metastasis. More than 3 cycles of chemotherapy were administered to treat all cases. The responses to chemotherapy were as follows: complete response, 1 case; partial responses, 13 cases; stable disease, 6 cases; and progressive disease, 5 cases (resistance was acquired after an initial good response). All cases achieved complete resection. 

After neoadjuvant chemotherapy, ctDNA was detected in 4 (16%) of 25 R0 resection cases, of which 3 cases were cases of liver metastasis and one case was a case of lung metastasis. Three of 4 cases acquired resistance and the disease progressed before metastasectomy. A good response to chemotherapy was observed and ctDNA was not detected in all, except 2 (#11 and #51b), of the remaining 21 cases. After metastasectomy, no ctDNA-positive case was found (Table 4).

### 3.6. Group 4: Neoadjuvant Chemotherapy Followed by Metastasectomy (R0 Resection Failed = 11) 

This group of patients received chemotherapy, followed by metastasectomy; however, R0 resection failed. Nine of 11 cases had synchronous metastasis. Most cases (8 of 11 cases) showed poor response to chemotherapy. Three (27.3%) cases were found to be ctDNA positive before liver metastasectomy or peritonectomy, with tumors measuring >1 cm and disease progression at the time of ctDNA sampling. After metastasectomy, ctDNA was detected in 4 (36.4%) cases with rapid progressive disease in the liver or in the liver and bone (Table 5). 

### 3.7. Longitudinal Tracking with Serial ctDNA Analysis

We conducted longitudinal ctDNA tracking in 14 cases. During serial sampling, the VAF in ctDNA increased before metastasis was radiologically confirmed in the case of recurrence (case number 6, 8, 20, 25, 26, 37, 40, 51, 53, 61, 65, 79, 92; all cases are listed in Appendix A); however, ctDNA was not detected after local interventions such as repeated surgery or radiofrequency ablation for metastatic lesions (case number 6, 20, 25, 37, 40, 51, 79, 92).

Moreover, although gross lesions are present, ctDNA might not be detected if the effective chemotherapy and radiotherapy are continued. In case number 26, ctDNA was not detected during maintenance chemotherapy for lung metastasis, but when a new brain metastasis occurred, ctDNA was detected. An increase in the VAF in ctDNA during treatment is highly associated with tumor growth and is detectable postmetastasectomy.

In case number 19, advanced gastric cancer occurred after 4.5 years of the diagnosis of mCRC with a novel pathogenic variant (TP53) from the gastric cancer tissue, which was not found in colon cancer tissue. During follow-up after gastrectomy, this pathogenic variant was repeatedly detected in the blood, confirming the recurrence of gastric cancer in the supraclavicular lymph nodes.

The detection rate of ctDNA differed based on the size and location of the metastatic lesion. Among R0 resection cases, the detection of ctDNA was less in small tumors (<1 cm) and lung metastasis and was high in large tumors (≥1 cm) and liver metastasis (Table 6; Figure 2). As demonstrated in this study, biological factors such as metastatic milieu shedding ctDNA other than tumor burden should be considered. In case of concurrent chemoradiotherapy or neoadjuvant chemotherapy for primary tumor, the VAF in the tumor was extremely low or under the detection level. This could be related to the lack of ctDNA detection in group 1 (case number 2, 7, 34, 40, 58, 60, 68, 72). Although not statistically presented, ctDNA was detected in several cases before several months of bone metastasis (case number 6) or confirmation of disease progression with radiological imaging (case number 40, 53).

## 4. Discussion

The optimal treatment strategies for mCRC differ based on the metastatic patterns. In patients with resectable and synchronous metastasis, surgical resection can offer a clear and significant long-term survival benefit. However, in patients with synchronous, extensive, large-volume, and disseminated disease, systemic chemotherapy is preferred and metastasectomy may be considered depending on the response to chemotherapy. No definite biomarker is available whether chemotherapy should be performed or the effectiveness of therapy regimen after successful R0 resection. 

Numerous reports have demonstrated that the baseline ctDNA level is a prognostic factor in a wide range of patients with metastatic cancers undergoing chemotherapy and sensitive assessment for early response monitoring [21,22,23]. ctDNA levels are associated with tumor burden. However, other factors influencing ctDNA levels are poorly characterized. Additionally, limited studies have reported data specifically on the use of ctDNA in patients with mCRC who have undergone metastasectomy. 

To determine whether the presence of ctDNA before and after metastasectomy can be a prognostic or predictive biomarker that aids in determining the treatment direction, we used a personalized approach for tumor-guided ctDNA detection. Nearly one-third of the samples were excluded from the analysis because of low yield. Among the 98 metastasectomies, 10 failed quality control and 18 did not have clinically significant pathogenic variants. ctDNA was detected in 16 (24%) of 67 events before metastasectomy; 9 (29%) of 31 cases treated with upfront metastasectomy without chemotherapy (groups 1 and 2) and 7 (19.4%) of 36 cases treated with neoadjuvant chemotherapy followed by metastasectomy (groups 3 and 4). After metastasectomy, ctDNA was detected in 4 (6%) of 67 events, all of which were found in group 4, and these were patients with gross lesions after metastasectomy. Compare with postoperative ctDNA analysis in the stage l to lll CRC, ctDNA-positive events were 10(8%) out of 125 patients at postoperative day 30 [24].

Further, we evaluated whether administration of chemotherapy was not statistically significantly related to ctDNA positivity before metastasectomy; the ctDNA detection rate tended to be low in cases of good response to chemotherapy. The detection of ctDNA rate was higher in liver metastasis and metastasized tumors measuring ≥1 cm than in lung metastasis and metastasized tumors measuring <1 cm. ctDNA was detected in 4 cases immediately after metastasectomy. All the 4 cases belonged to group 4, which underwent metastasectomy (failed R0 resection) with neoadjuvant chemotherapy.

In our study, the ctDNA detection rate was lower than that reported in a previous study on mCRC [9]. Considering the ctDNA dynamics corresponding to the clinical condition of patients with mCRC [25,26], most patients’ tumor burden was low and chemotherapy response was the best at the time of ctDNA blood sampling. The mean diameter of the resected tumor was 2.7 cm in ctDNA-positive cases and 1.4 cm in ctDNA-negative cases. Primary tumor burdens of 1, 10, and 100 cm^3^ result in mean clonal plasma VAFs of 0.008%, 0.1%, and 1.4%, respectively [27]. In a previous study, considering the ctDNA shedding rate per cancer cell, the probabilities of a false-negative result for a particular actionable mutation clonally present in tumors with diameters of 1, 1.5, and 2 cm were 82%, 44%, and 9.3%, respectively [28]. In the case of patients with small, isolated metastatic tumors who did not receive chemotherapy (groups 1 and 2), the median VAFs in the tumor and plasma ctDNA were 28.7% and 1.1%, respectively, which the detection rate in plasma ctDNA can be less than one-tenth of that of tumor DNA. The difference of the above frequencies could be partially explained by the vulnerability of ctDNA. According to the studies on early dynamics of ctDNA, changes in ctDNA mutation frequencies were generally observed during the first 2 cycles of chemotherapy [29,30]. This is the reason that the neoadjuvant chemotherapy groups (groups 3 and 4) showed a lower ctDNA positive rate than the upfront metastasectomy groups (groups 1 and 2). ctDNA is typically undetectable in mid-treatment samples even in patients with metastatic cancer with measurable disease on imaging [31].

To the best of our knowledge, this is the first prospective study to investigate ctDNA levels before and after metastasectomy in patients with potentially resectable mCRC. We aimed to detect ctDNA before and after metastasectomy with the same detection method and identify the clinicopathological determinants of ctDNA detection. The released ctDNA may be influenced by various mechanisms such as the metastatic tumor site, invasion through blood vessels, administration of effective local and systemic chemotherapy, and tumor burden. 

This study has several limitations. The relatively small sample size of this exploratory study impairs the generalization of our conclusion. Further, we did not collect serial ctDNA samples from all patients. Therefore, the interpretation of serial ctDNA tracking is limited. There is also the potential that our ctDNA sequencing assay was not sensitive enough to detect the true landscape of plasma mutations and molecular alterations. Finally, our study was not powered to examine the association between ctDNA change and clinical outcomes such as PFS and OS. Further studies should include the systemic sample collection, a larger sample to allow clinical factors associated with prognosis to be controlled, such as receiving neoadjuvant chemotherapy before metastasectomy, components of the clinical risk factors, and R0 resection status. 

## 5. Conclusions

After metastasectomy for oligometastatic lesions and a good response to chemotherapy, most of the ctDNA was cleared and existed below the detection level. The biological characteristics affecting the release of tumor DNA should be considered when applying ctDNA assays in a clinical setting. Unexpectedly low fractions of ctDNA were found in a substantial proportion of patients with metastatic disease [32]. To expand the clinical application of ctDNA as a biomarker, it would be more cost-effective to follow up specific tumor-derived ctDNA with a highly sensitive tool such as droplet digital PCR rather than whole exome or targeted sequencing in the context of clinical setting.

## Figures and Tables

**Figure 1 cancers-13-02231-f001:**
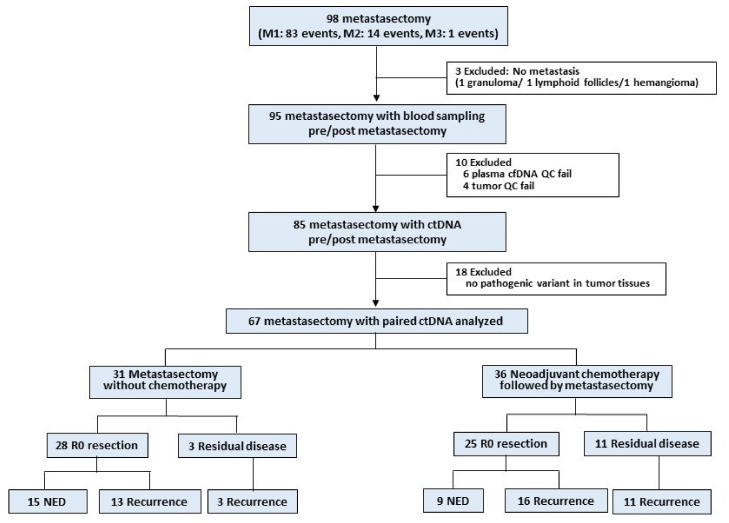
Workflow of the exploratory prospective study.

**Figure 2 cancers-13-02231-f002:**
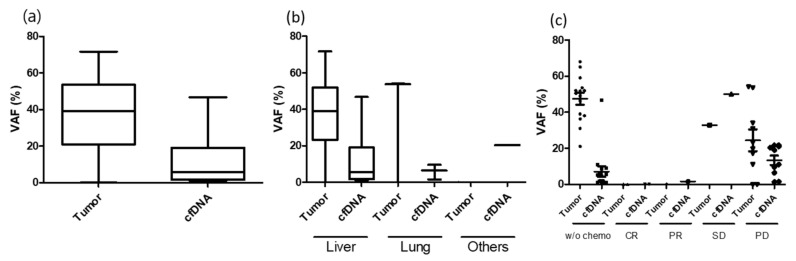
VAF difference between tumor and plasma under various conditions. (**a**) Mean VAF difference between tumor and plasma; (**b**) VAF difference between tumor and plasma under metastatic organ; (**c**) VAF difference between tumor and plasma under neoaduvant chemotherapy response.

**Table 1 cancers-13-02231-t001:** Patient Characteristics.

Clinical Variables		*N*_patient_ = 58 (%)
Age years (median, range)		56 (35–78)
Gender	Male	37 (63.8)
	Female	21 (36.2)
Primary tumor site	Ascending colon	8 (13.8)
	Descending colon	20 (34.5)
	Rectum	30 (51.7)
Primary tumor surgery	Yes	57 (98.3%)
	No	1 (1.7%)
Synchronicity of metastasis	Metachronous	18 (31.1)
	Synchronous	40 (68.9)
Metastasectomy organ	Liver	38 (65.5)
	Lung	25 (43.1)
	Peritoneum	4 (6.9)
	Lymph node	2 (3.4)

**Table 2 cancers-13-02231-t002:** Metastasectomy (R0 resection) Without Chemotherapy (*N*_event_ = 28).

No	Synchronicity	Variant	Primary Tissue VAF		Metastatic Tissue VAF	Metastatic Site	PreM ctDNA VAF	PostM ctDNA VAF	Recurence
			No CTx	CTx					
ctDNA positive before metastasectomy									
020b	M	TP53 R196 *		38.5	65.1	Liver 2.5 cm	1.2	0.0	N
		APC Q1294 *		25.4	51.6		0.0	0.0	
037b	M	TP53 T211I	14.9		59.1	Liver 2.5 cm	46.8	0.0	N
073	M	APC T1445Qfs	21.0		26.0	Liver 7.4 cm	0.0	0.0	N
		APC R876 *	31.2		0.0		0.0	0.0	
		TP53 R248W	23.0		21.0		1.1	0.0	
025a	M	FXW7 R465C	33.0		33.4	Liver 2.1 cm	0.0	0.0	N
		KRAS G12D	23.8		28.4		0.0	0.0	
		TP53 R196 *	35.9		51.8		2.1	0.0	
		APC L1488fs	26.1		0		0.0	0.0	
025b	M	FXW7 R465C	33.0		31.0	Liver 1.5 cm	1.1	0.0	Liver LN
		KRAS G12D	23.8		36.3		2.0	0.0	
		TP53 R196 *	35.9		49.0		1.8	0.0	
		APC L1488fs	26.1		0		0.0	0.0	
050	M	APC Q1367 *	32.6		34.0	Liver 3.0 cm	0.0	0.0	Liver
		TP53 R237H	26.2		39.0		1.0	0.0	
		BRAF G469E	0.0		1.3		0.0	0.0	
		CTNNB1 G34E	0.0		1.4		0.0	0.0	
		VHL R161 *	0.0		1.4		0.0	0.0	
062	M	APC R1450 *	35.3		52.0	Liver 3.2 cm	5.9	0.0	Lung
		KRAS G12S	55.0		68.0		8.6	0.0	
		TP53 R342 *	50.9		52.0		9.0	0.0	
092	M	SMAD4 G419R	QC failed *		53.6	Liver 4.7 cm	11.0	0.0	Lung Liver
ctDNA negative before metastasectomy									
002	M	NRAS G12D		0.0	19.0	Lung 0.9 cm	0.0	0.0	N
005	M	TP53 R282W	43.0		22.0	Lung 0.8 cm	0.0	0.0	N
		APC R876 *	34.0		14.0		0.0	0.0	
028b	M	KRAS G12V	14.3		0.0	PT 4.0 cm	0.0	0.0	N
		PIK3CA E545K	14.8		0.0		0.0	0.0	
		SMAD4 R361H	16.4		0.0		0.0	0.0	
		SMAD4 E330Q	16.6		50.2		0.0	0.0	
		PIK3CA G914R	14.9		0		0.0	0.0	
034	M	TP53 R175H		0.0	17.2	LN 0.9 cm	0.0	0.0	N
		KRAS G12C		0.0	18.0		0.0	0.0	
		PIK3CA E545V		0.0	3.3		0.0	0.0	
041	M	KRAS A59T	59.0		44.0	Lung 1.0 cm	0.0	0.0	N
		TP53 R342 *	44.0		22.0		0.0	0.0	
056	M	KRAS G12D	24.0		0.0	Liver 0.7 cm	0.0	0.0	N
		TP53 C238Y	35.7		0.0		0.0	0.0	
060	M	TP53 N239 *		21.0	32.0	Liver 2.0 cm	0.0	0.0	N
066	M	TP53 R282W	23.4		3.9	Lung 0.4 cm	0.0	0.0	N
		APC R876 *	31.2		0.0		0.0	0.0	
		APC E1379 *	26.4		0.0		0.0	0.0	
068	M	TP53 R213 *		0.0	25.1	Lung 1.7 cm	0.0	0.0	N
081	S	KRAS G12S	30.0		29.0	Lung 1.2 cm	0.0	0.0	N
		APC Q1378 *	23.0		25.0		0.0	0.0	
093	M	PIK3CA V344M	BDL *		17.4	Lung 1.4 cm	0.0	0.0	N
098	M	KRAS G13D	0.0		21.1	Liver 2.2 cm	0.0	0.0	N
		APC L1488fs	22.9		0.0		0.0	0.0	
		APC E1494fs	0.0		20.4		0.0	0.0	
007	M	APC C1289 *		6.9	20.8	Lung 1.4 cm	0.0	0.0	PT
		PIK3CA E542K		3.8	20.5		0.0	0.0	
		TP53 R282W		2.6	14.0		0.0	0.0	
		TP53 S99fs *		2.1	11.0		0.0	0.0	
031a	S	APC R1114 *		8.1	29.4	Lung 1.4 cm	0.0	0.0	Lung
		KRAS G13C		13.4	42.6		0.0	0.0	
		APC R1463fs		2.0	0.0		0.0	0.0	
		APC E1345 *		8.1	30.7		0.0	0.0	
032	M	KRAS G12D	22.3		29.2	Lung 1.2 cm	0.0	0.0	LN
		TP53 R282W	0.0		34.6		0.0	0.0	
		TP53 S127F	14.3		0.0		0.0	0.0	
		APC L1488fs	13.2		0.0		0.0	0.0	
035	M	KRAS G12C	37.7		15.6	Lung 0.5 cm	0.0	0.0	Lung
		TP53 I195T	41.4		17.1		0.0	0.0	
		APC R1463fs	1.9		0.0		0.0	0.0	
		APC S1501fs	24.0		8.9		0.0	0.0	
040	M	KRAS G12D		3.4	18.5	Lung 0.6 cm	0.0	0.0	Lung Bone
		TP53 S241P		1.2	10.4		0.0	0.0	
		PTEN R173H		1.0	0.0		0.0	0.0	
058	M	NRAS G12D		24.6	32.7	Lung 1.0 cm	0.0	0.0	Lung
		TP53 H297fs		26.2	14.6		0.0	0.0	
		TP53 R333fs		22.5	10.0		0.0	0.0	
072	M	APC R876 *		17.8	10.6	PT 2.7 cm	0.0	0.0	PT LN
		TP53 R248Q		16.8	10.8		0.0	0.0	
		APC Q1303 *		14.1	12.0		0.0	0.0	
078	M	KRAS G12V	41.0		12.9	Lung 1.2 cm	0.0	0.0	Lung
		TP53 R175H	54.1		20.8		0.0	0.0	

Abbreviations: VAF, variant allele frequency; PreM, premetastasectomy; PostM, postmetastasectomy; CTx, chemotherapy; M, metachronous stage 4; S, synchronous stage 4; N, no recurrence; PT, peritoneum; LN, lymph node; BDL, below detection level. QC failed *: this patient received concurrent chemoradiation followed by surgery; *, indicate a translation termination (stop) codon.

**Table 3 cancers-13-02231-t003:** Metastasectomy without Chemotherapy and Residual Disease (*N*_event_ = 3).

No	Synchronicity	Variant	Primary Tissue VAF		Metastatic Tissue VAF	Metastatic Site	PreM ctDNA VAF	PostM ctDNA VAF	Residual
			No CTx	CTx					
ctDNA positive before metastasectomy									
046	M	KRAS G12D		0	38.1	Liver 2.1 cm	4.4	0.0	Lung <0.5 #4 Pelvic LNs
		TP53 G245C		1.0	46.3		5.6	0.0	
		APC K1561 *		0	50.5		5.0	0.0	
ctDNA negative before metastasectomy									
022	M	TP53 V272M	39.4		55.4	Liver 3.0 cm	0.0	0.0	LN 2.2 cm
		SMAD4	7.4		0.0				
061c	M	TP53		51.5	25.1	Lung 2.1 cm	0.0	0.0	PT
		APC H1490fs		30.2	16.2		0.0	0.0	

Abbreviations: VAF, variant allele frequency; PreM, premetastasectomy; PostM, postmetastasectomy; CTx, chemotherapy; M, metachronous stage 4; PT, peritoneum; LN, lymph node; *, indicate a translation termination (stop) codon.

**Table 4 cancers-13-02231-t004:** Neoadjuvant Chemotherapy Followed by Metastasectomy (R0 resection) (*N*_event_ = 25).

No	Synchronicity	Variant	Primary Tissue VAF			Metastatic Tissue VAF		Meta Site	PreM ctDNA	PostM ctDNA	Recurence
			No CTx	CTx	Rx	CTx	Rx				
ctNDA positive before metastasectomy											
001	S	TP53 I255F		49.8	SD	0.0	PR	Liver 0.1 cm	1.6	0.0	N
036	S	APC R1114 *	44.2			31.0	PD	Liver 2.5 cm	21.0	0.0	N
		APC R1450 *	21.5			17.0			11.0	0.0	
		KRAS G12D	46.5			34.0			19.0	0.0	
008b	S	KRAS Q61R	48.8			53.7	PD	Lung 2.5 cm	9.6	0.0	Lung
		TP53 R175H	49.5			54.1			6.4	0.0	
028a	S	KRAS G12V	14.3			7.8	PD	Liver 2.2 cm	0.0	0.0	PT
		PIK3CA E545K	14.8			11. 1			1.3	0.0	
		PIK3CA G914R	14.9			10.2			0.0	0.0	
		SMAD4 E330Q	16.6			8.9			0.0	0.0	
		SMAD4 R361H/C	16.4			1.7			0.0	0.0	
ctNDA negative before metastasectomy											
004	S	TP53 R273C	60.4			4.1	PR	Liver 0.8 cm	0.0	0.0	N
		APC K1308 *	40.2			2.8			0.0	0.0	
012	S	KRAS G12V	14.6			0.0	CR	Liver 0 cm	0.0	0.0	N
		PIK3CA H1047R	20.6			0.0			0.0	0.0	
031b	S	APC R1114 *	8.1			11.6	SD	Lung 1.4 cm	0.0	0.0	N
		APC E1345 *	8.1			15.4			0.0	0.0	
		KRAS G13C	13.4			9.1			0.0	0.0	
042	M	APC Q1378 *	47.5			36.2	PR	Liver 2.5 cm	0.0	0.0	N
		KRAS G12D	29.9			18.3			0.0	0.0	
		TP53 R175H	30.5			19.3			0.0	0.0	
044	S	CTNNB1 S45F	36.8			6.5	PR	Liver 2.2 cm	0.0	0.0	N
		KRAS G12D	17.3			0.0			0.0	0.0	
		PTEN R335 *	5.2			0.0			0.0	0.0	
057	S	TP53 C275Y	26.5			36.9	SD	Lung 1.6 cm	0.0	0.0	N
091	S	APC E1306 *	18.0			0.0	PR	Liver 0.8 cm	0.0	0.0	N
		TP53 R175H	40.2			6.2			0.0	0.0	
		APC Q886 *	26.9			4.6			0.0	0.0	
006	S	KRAS G13C	37.2			10.7	PR	Liver 1.6 cm	0.0	0.0	Bone
		TP53 R306 *	34.9			9.1			0.0	0.0	
011	M	APC R876 *	30.6			0.0	PD	Lung 1.0 cm	0.0	0.0	Lung
		KRAS E545K	34.3			2.2			0.0	0.0	
		TP53 R342 *	0.0			2.5			0.0	0.0	
		SMAD4 R361H	0.0			3.8			0.0	0.0	
		SMAD4 A118V	64.8			0.0			0.0	0.0	
014	S	KRAS Q61H	25.3			5.6	SD	Liver 2.3 cm	0.0	0.0	PT
		SMAD4 R361C	15.2			3.0			0.0	0.0	
019	S	PIK3CA G1049R		3.1	SD	3.6	SD	Liver 0.9 cm	0.0	0.0	Liver
020a	S	TP53 R196 *		38.5	SD	21.2	PR	Liver 2.5 cm	0.0	0.0	Liver
		APC Q1294 *		25.4		13.8			0.0	0.0	
027	S	TP53 R175H		6.1	PR	2.6	PR	Liver 3.2 cm	0.0	0.0	Liver Lung
		PTEN R335 *		1.1		0.0			0.0	0.0	
037a	S	TP53 T211I	14.9			1.0	PR	Liver 0.5 cm	0.0	0.0	Liver
048	S	APC R1450 *		1.5		0.0	PR	Lung 0.6 cm	0.0	0.0	Lung
		CTNNB1 S37Y		0.0		21.2			0.0	0.0	
		KRAS G12D		1.4		0.0			0.0	0.0	
		PIK3CA E545K		2.0		0.0			0.0	0.0	
		TP53 R209Kfs		1.8		0			0.0	0.0	
051b	M	KRAS G13S	0.0			1.1	PD	Lung 0.9 cm	0.0	0.0	Lung
		PTEN R130Q	0.0			1.0			0.0	0.0	
		SMAD4 C115Y	0.0			1.2			0.0	0.0	
		TP53 R175H	10.7			40.0			0.0	0.0	
052	S	RB1 A201fs		0.0		5.5	PR	Lung 0.4 cm	0.0	0.0	Lung
055	S	KRAS G13D	60.3			16.6	PR	Liver 3.5 cm	0.0	0.0	Liver
061b	S	TP53?		51.5	SD	27.1	SD	Lung 0.5 cm	0.0	0.0	Lung
		APC H1490fs		30.2		19.5			0.0	0.0	
063	S	KRAS G12V	BDL			5.3	PR	Liver 2.2 cm	0.0	0.0	Lung
		KRAS A146T	BDL			25.0			0.0	0.0	
		TP53 V173L	BDL			2.4			0.0	0.0	
		APC P1381fs	BDL			15.5			0.0	0.0	
067	S	TP53 M246R		23.2	SD	39.3	SD	Rec 3.4 cm	0.0	0.0	PT

Abbreviations: VAF, variant allele frequency; PreM, premetastasectomy; PostM, postmetastasectomy; CTx, chemotherapy; Rx, chemotherapy response; CR, complete response; PR, partial response; SD, stable disease; PD, progressive disease; M, metachronous stage 4; S, synchronous stage 4; N, no recurrence; PT, peritoneum; LN, lymph node; Rec, rectum; BDL, below detection level; *, indicate a translation termination (stop) codon.

**Table 5 cancers-13-02231-t005:** Neoadjuvant Chemotherapy Followed by Metastasectomy and Residual Disease (*N*_event_ = 11).

No	Synchronicity	Variant	Primary Tissue VAF			Metastatic Tissue VAF		Meta Site	PreM ctDNA VAF	PostM ctDNA VAF	Residual
			No CTx	CTx	Rx	CTx	Rx				
ctDNA positive before metastasectomy											
053	S	EBRR2 D769Y	0			71.7	PD	Liver #2 (2.3)	32.9	50	Liver multiple Bone
076	S	TP53 I195fs	19.2			0	PD	PT 4.1 cm	20.4	16.9	Huge liver
079	M	APC Q1378 *	BDL			23.2	PD	Liver #7	21.6	0	Lung 0.9 cm
		TP53 S215G	BDL			19.8			21.7	0	
096	S	KRAS G13D	31.4			8.6	PR	Liver #9	0	0	Liver #2 (<0.5)
		TP53 V73fs	50.8			0			0	0	
ctDNA negative before metastasectomy											
003	S	TP53 R196 *	30.2			22.8	PD	Liver 2.3 cm	0	0	Liver 1.6 cm
008a	S	KRAS Q61R	48.8			10.4	SD	Lung 0.6 cm	0 0	0 0	Lung 0.8 cm
		TP53 R175H	49.5			10.1					
026	S	TP53 R306 *	39.7			25.2	PD	Liver #5 PT #5	0	0	Liver 0.5 cm
051a	S	TP53 R175H	10.7			11.5	PD	Liver #4	0	0	Lung 0.5 cm
		BRAF V471F	0			12.3			0	0	
054	M	KRAS G12D	45.5			49.9	PD	Liver 1.6 cm PT 2.6 cm	0	10.1	Liver #6 (<1.0 cm)
061a	S	TP53?		51.5	SD	14.5	PD	Lung 1.0 cm	0	0	Lung <0.5 cm #2
		APC H1490fs		30.2		12.4			0	0	
065	S	APC R1114 *	37.5			37.5	PR	Liver #4	0	3.6	Liver 1.4 cm
		TP53 G244S	52.7			46.2			0	2.2	
		APC S1356 *	29.3			15.8			0	2.9	

Abbreviations: VAF, variant allele frequency; PreM, premetastasectomy; PostM, postmetastasectomy; CTx, chemotherapy; Rx, chemotherapy response; PR, partial response; SD, stable disease; PD, progressive disease; M, metachronous stage 4; S, synchronous stage 4; N, no recurrence; PT, peritoneum; LN, lymph node; Rec, rectum; BDL, below detection level; *, indicate a translation termination (stop) codon.

**Table 6 cancers-13-02231-t006:** ctDNA Detection before Metastasectomy (*N*_events_ = 67).

Clinical Condition		ctDNA Positive (*N* = 16)	ctDNA Negative (*N* = 51)	*p* Value
Neoadjuvant chemotherapy before R0 resection	Yes	7	29	0.3587
	No	9	22	
Metastasectomy organ	Liver	14	21	0.0045
	Lung	1	24	
	Other	1	6	
Tumor burden (tumor diameter)	>1 cm	15	28	0.0183
	≤1 cm	1	23	

## Data Availability

The data presented in this study are available on request from the corresponding author. The data are not publicly available due to a privacy issue from the patients.

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
