# Peer review of "Clinical Implication of Liquid Biopsy in Colorectal Cancer Patients Treated with Metastasectomy"

_cancers, 2021, doi:10.3390/cancers13092231_

Round 1

Reviewer 1 Report

Relevant research on biomarker for  colorectal metastasis .The authors have highlighted the limitation of the study.

It would  have been more informative if the whole spectrum of circulating Tumour DNA for each patient from diagnosis of the metastatic  CRC before long course Chemorad or primary resection to treatment of primary CRC to metastesctomy and after  metastasectomy. I think the most useful information to glen from CtDNA is to answer the question of missed metastasis in CRC which is about 20-25%. The role of ctDNA in CRC surveillance after metastasectomy who have been useful.

I want to believe the study set out to assess the Progression-free survival (PFS) was measured from the time of metastasectomy to recurrence or death from disease- I can't see this information

I want to believe that the study also set out  to assess -Overall survival (OS) was measured from 141 the time of metastasectomy to death from the disease. - I can't see this information

Furthermore , it would have also bemore  informative to know the ctDNA  of patient with recurrence CRC following metastasectomy and patient who were unable to have primary treatment of the disease due to locally advance disease with wide spread metastasis which may be used in assessing response to chemotherapy if t ctDNA levels were undetectable after removal of metastasectomy following neoadjuvant chemorad

What is the rationale assessing Ct DNA 3-4 weeks after metastasectomy, are there evidence to suggest that this circulating tumour cell are cleared the system after 4 weeks.

I think one of the major flaw of the study  which reflected in the large drop out rate was the  poor yield from paraffin embedded  tissue for  CtDNA extraction which was not discussed in the manuscript

Author Response

It would have been more informative if the whole spectrum of circulating Tumour DNA for each patient from diagnosis of the metastatic CRC before long course Chemorad or primary resection to treatment of primary CRC to metastesctomy and after metastasectomy. I think the most useful information to glen from CtDNA is to answer the question of missed metastasis in CRC which is about 20-25%. The role of ctDNA in CRC surveillance after metastasectomy who have been useful.

Ans> You’re right. I agree that I would have serially followed up ctDNA from diagnosis of CRC via metastasis to surgery followed by chemotherapy. Unfortunately my original hypothesis was that  postoperative ctDNA could guide adjuvant chemotherapy incorporation after metastasectomy. If ctDNA positive after metastasectomy, chemotherapy would be more helpful to prevent recurrence. That’s the reason why I just select two times of blood sampling before and after metastasectomy. I just collected some patients with exploratory aims.  

I want to believe the study set out to assess the Progression-free survival (PFS) was measured from the time of metastasectomy to recurrence or death from disease- I can't see this information

I want to believe that the study also set out  to assess -Overall survival (OS) was measured from 141 the time of metastasectomy to death from the disease. - I can't see this information

Ans> All survival data (PFS and OS) were updated in supplementary table 2.

I didn’t present ctDNA positivity and survival outcomes in the body because ctDNA positivity before and after metastasectomy had no significant effect on PFS and OS.

Furthermore, it would have also be more informative to know the ctDNA of patient with recurrence CRC following metastasectomy and patient who were unable to have primary treatment of the disease due to locally advance disease with wide spread metastasis which may be used in assessing response to chemotherapy if t ctDNA levels were undetectable after removal of metastasectomy following neoadjuvant chemorad

Ans> I agreed and accepted your critics.

As I mentioned at your first question, I originally didn’t plan to follow up serial ctDNA collection in all patients after metastasectomy. For effective ctDNA research, it is necessary to trace the disease trajectory of mCRC.

My original idea was to see if ctDNA would be possible as a tool to select patients who need additional chemotherapy after metastasectomy. I excluded the patient who were unable to have primary treatment of the CRC. I agreed this would be limits of my research.

What is the rationale assessing Ct DNA 3-4 weeks after metastasectomy, are there evidence to suggest that this circulating tumour cell are cleared the system after 4 weeks.

Ans> The postoperative chemotherapy used to start 3-4 weeks after surgery. That’s the reason why I decided ctDNA collecting time point at the research ideation. I hoped that patients who truly needed chemotherapy could be selected based on ctDNA positivity. I couldn’t imagine ctDNA vulnerability and releasing condition. Thus I just follow up Reiner’s work (newly added as reference number 24) which collected ctDNA at postoperative 30 days. To be honest, there was no strong evidence of my post-surgical ctDNA collecting time point.  

I think one of the major flaw of the study which reflected in the large drop out rate was the poor yield from paraffin embedded tissue for ctDNA extraction which was not discussed in the manuscript

Ans> Among 31 of 98 metastatic events, 10 cases showed QC failure. Those would be related with paraffin embedded tissue limitation.

In consideration of clinical application, I only selected pathogenic variants which exist in tumor and plasma at the same time. Therefore I excluded all non-pathogenic variants and if pathogenic variants were found only in either tumor or plasma. These selection criteria would be more related with high drop rate of this study population.

Reviewer 2 Report

The main question is the clinical relevance of such observation.

  1. How technical condition for liquide biopsy were determined in comparison to the litterature.
  2. The rate of positivity is very small. It is important to compare these results with currently litterature
  3. A figure to represent relation between VAF and postive read in blood vs tumor / Idem VAF in blood and tumor and size of tumor or response to previous line will improve the analysis.
  4.  Does blood ctDNA is prognostic of survival.

Author Response

How technical condition for liquid biopsy were determined in comparison to the literature.

As I quoted the liquid biopsy sequencing method as reference number 19 in the text, ultra-deep targeted sequencing of ctDNA was performed using Ion AmpliSeq Cancer Hotspot Panel v2 (ICP; Iron Torrent) and the Proton Platform; this panel covers 2,800 COSMIC mutations from 50 cancer genes. ICP results were validated with ddPCR in the previous study. The distribution of sequence lengths was between 60 and 170 bp. Targeted sequencing using the ICP panel generated approximately 604 Mb per sample with an average of 90.18% on target. Sequences of all samples achieved a mean depth of 25,804×.

The rate of positivity is very small. It is important to compare these results with currently literature

Based on your comment, I have clearly stated this on line 281 in the discussion part as below;

ctDNA was detected in 16 (24%) of 67 events before metastasectomy; 9(29%) of 31 cases treated with upfront metastasectomy without chemotherapy (groups 1 and 2) and 7(19.4%) of 36 cases treated with neoadjuvant chemotherapy followed by metastasectomy (groups 3 and 4). After metastasectomy, ctDNA was detected in 4(6%) of 67 events, all of which were found in group 4, and these were patients with gross lesions after metastasectomy. Compare with postoperative ctDNA analysis in the stage l to lll CRC, ctDNA-positive events were 10(8%) out of 125 patients at postoperative day 30.

A figure to represent relation between VAF and positive read in blood vs tumor / Idem VAF in blood and tumor and size of tumor or response to previous line will improve the analysis.

Thank you for your valuable comments. I added a figure which compared tumor and plasma ctDNA VAF as Figure 2 in the discussion part.

Does blood ctDNA is prognostic of survival?

Based on my study result, it’s not. The rate of ctDNA detection before metastasectomy was 16 (24%) of 67 events before metastasectomy and 4 (6%) after metastasectomy. Surgical skill and chemotherapy after surgery could be confounding factors. Post-metastasectomy ctDNA could be the prognostic biomarker. It needs more sensitive assay and schematic serial follow up.

Reviewer 3 Report

The authors collected and analyzed the ctDNA detection rate in patients who underwent metastatectomies from mCRC. They found that the detection rate was higher in liver metastasis than lung metastasis, and it was also higher in tumors measuring > 1cm than tumors <1cm. The detection level of ctDNA was mostly cleared or extremely low in those with good response to chemotherapy.

The originality and strength of this study was the evaluation of ctDNA before and after metastatectomy in patients with potentially resectable mCRC, although there may be some arguments regarding the number of patients, heterogeneity in studied population, availability of sample data, and the detection rate of ctDNA in this study.

Despite the aforementioned limitations, this paper may promote further studies on the clinical significance of measuring sequential ctDNA in patients with potentially resectable mCRC in order to launch a tailored treatment strategy.

Grouping analysis based on the neoadjuvant chemotherapy and R0 resection was very interesting, so hopefully they will continue investigating this subject and publish their data with more comprehensive dataset with larger cohort.

Author Response

Dear Reviewer,

Thank you for your valuable comments and acknowledging. In order to find biomarkers that can prevent recurrence after liver metastasectomy, I’ll conduct more comprehensive studies with by integrating various liquid biopsy techniques including ctDNA.

Round 2

Reviewer 1 Report

Good piece of research.

Reviewer 2 Report

no additional comments